# National Public Health Response to *Candida auris* in England

**DOI:** 10.3390/jof5040093

**Published:** 2019-10-03

**Authors:** Colin S. Brown, Rebecca Guy

**Affiliations:** 1Healthcare Associated Infections and Antimicrobial Resistance Division, National Infection Service, Public Health England, 61 Colindale Avenue, London NW9 5EQ, UK; 2Department of Infection, Royal Free Hospital, Pond Street, London NW3 2QG, UK

**Keywords:** emerging, response, development, public health, fungal, novel, engagement

## Abstract

This paper highlights the key steps undertaken by a national public health agency, working in close collaboration with academic partners and experienced healthcare professionals, in developing a response to the rapid emergence of a novel nosocomial pathogen. It details the key activities in national incident management team formation, surveillance activities, epidemiological investigations, laboratory developments, scientific advances, and collaborative activities. It discusses commonalities that can be adapted for dealing with the emergence of future new pathogens.

## 1. Introduction 

*Candida auris* is a rapidly emerging global pathogen with five distinct clades characterised across separate continents over the past ascribed to the geographical region from which they were first described—South American, South African, South Asian, East Asian and a putative Iranian clade [1]. It has brought significant challenges to the infection control world as the first fungal agent to cause large, difficult to control nosocomial outbreaks [2]. There continue to be global reports of prolonged hospital outbreaks, and year-on-year new countries report cases of infection for the first time. Considerable questions remain regarding its simultaneous origin, optimum disinfection regimens, methods for decolonisation or deburdening, treatment options, community prevalence, modes of spread, and prevention of spread of the of multi-drug resistant disease. Nonetheless, there has been significant progress made in raising awareness, developing hospital surveillance policies, developing organism culture, identification, and speciation laboratory methods and learning from international best practice [3]. It is highly likely that additional bacteria, fungi or other nosocomial pathogens will emerge due to selection pressures such as climate change, increasing patient immunosuppression, broad-spectrum antibiotic use and unknown factors that will influence new antimicrobial resistance profiles or indeed new pathogens. In this article, we will highlight the various multimodal research and development tools that national public health agencies will need to employ to be able to combat novel healthcare-associated infections, highlighting throughout the English experience of tackling *C. auris*.

## 2. Summary of Epidemiology in England

*C. auris* was first identified in England in 2013, with significant numbers detected through 2015 and particularly into 2016 [4]. Through the end of May 2019, Public Health England (PHE) has identified approximately 270 cases of *C. auris* in England, the majority of which are asymptomatic colonisations picked up through enhanced screening, but one quarter of which were symptomatic infections, including 35 patients with blood stream infections. There have been three large hospital outbreaks, all three declared over by the end of 2017, though each was prolonged and difficult to control, and two of which have been reported upon in depth [5,6]. There has been no known mortality directly attributable to *C. auris* infection in England.

There continue to be sporadic case introductions into English hospitals; for example, seven new hospitals have seen new case detections in the first five months of 2019. Many of these involved patients repatriated from international hospitals (including India, Qatar, Kuwait, Oman, Pakistan, and Kenya). A small number of hospitals that have seen previous cases of patients colonised with *C. auris* have reported sporadic new case detections. One hospital has seen two repeat introductions with transmission events. An additional three hospitals have had transmission events reported (all fewer than five cases), and one hospital with a previously reported outbreak has had further, though smaller, transmission events (thought to be associated with a new introduction). The transmission chains identified in 2018 and 2019 were limited due to early recognition, appropriate case isolation, enhanced infection prevention and control measures, and wider screening. One hospital has detected three distinct clades of *C. auris* in separate patients (all except the South American clade have been detected in England).

What is clear is that there has been multiple repeat introductions of *C. auris* via global transmission routes, both to the country as a whole, and to individual hospitals. As such, mechanisms to both detect and rapidly respond to such introductions must be designed and implemented. 

## 3. The Anatomy of a National Outbreak Response

Key to the development of a national response that encompassed clinical patient management, infection prevention and control (IPC) messaging, and public health approaches, was the convening of a national incident management team (IMT) in June 2016 with representatives from a diverse range of specialties and experience. This included hospital microbiologists and IPC professionals with direct experience of managing the new pathogen, communications and health protection team-members, infectious disease specialists, and experts in public health and environmental microbiology, reference mycology, and epidemiology. Following recognition of the first nosocomial outbreak in Europe [5], lookback exercises identified other hospitals with likely transmission.

National guidance was collaboratively produced [7] and widely disseminated. National experts in healthcare-associated infection (HCAI) public health and environmental microbiology performed site visits in hospital outbreak settings. Outputs were built on existing public health mechanisms, with key activities around surveillance, diagnostic development and scientific advances. Reactive needs were addressed iteratively, such as expanded guidance documents created to include advice on community care settings and patient transfer [8].

Regular bi-monthly meetings were initially held, which decreased in frequency as experience grew, documentation was developed and monitoring mechanisms were introduced. 

### 3.1. Surveillance Activities

Cognisant of the pitfalls in routine data collection for an emerging organism with no mandatory reporting, PHE developed a triangulated approach to data reporting. Firstly, laboratories with the capacity to detect *C. auris* feed their routine output into our national laboratory reporting for infectious diseases surveillance and antimicrobial resistance (Second Generation Surveillance System, SGSS). This is routinely monitored by a central team at the HCAI and Antimicrobial Resistance (AMR) Division of the National Infection Service. A laboratory questionnaire was conducted to ascertain what proportion of diagnostic laboratories have methods to identify *C. auris* locally (55% in 2017), and checks performed to ensure that this novel pathogen had an allocated field code that could be reported [9]. Secondly, all new isolates confirmed by the national Mycology Reference Laboratory (MRL), which includes 45% of laboratories unable to detect *C. auris* locally, are regularly fed back to the national surveillance team, with real-time alerts for new centres reporting *C. auris* detection. Thirdly, any report of a new case or request for advice received from local public health teams with a remit to investigate and manage health protection incidents (Health Protection Teams (HPTs)) is also passed onto the national team. In this manner, it is anticipated that all new cases of *C. auris* are captured so public health advice can be actioned. Standardised protocols for data collection to inform epidemiological and clinical risk assessment have been produced after piloting with HPTs.

### 3.2. Epidemiological Investigations

To further inform a risk assessment of the likelihood of multiple introductions and whether to consider recommendations for routine screening, a sentinel point prevalence survey was conducted on patient admission to eight English intensive care units (ICUs) that serve ethnically diverse populations (used as a proxy for travel to areas of higher endemicity reflective of the known global distribution of *C. auris*). All admissions were screened to provide estimates of prevalence on point of entry to the ICU. Preliminary results have been reported at international conferences [10], with no detection of *C. auris* detected in over 800 patient samples. Field Epidemiology Teams assisted with initial outbreak response to address key questions such as determining the highest positivity rate of different clinical screening sites, and assessing risk factors for case detection including prior antibiotic and antifungal administration. There remains on-going epidemiological data collection to better delineate risk factors for acquisition and progression to invasive infection, as well as interrogation on all-cause mortality for those with clinical infection (which remains lower than for candidaemias at approximately 20%) [11].

### 3.3. Laboratory Developments

With newly recognised pathogens, there will need to be significant investment in the development of both standard and reference laboratory techniques to rapidly. This is particularly true for *C. auris*, where traditional biochemical diagnostic methods can either provide no diagnosis, or misidentify species. Access to diagnostics for novel pathogens is key—all institutions unable to adequately differentiate *C. auris* from related species can send any suspect isolates to both the MRL, or a network of regional public health microbiology laboratories, all of which can detect *C. auris*.

The MRL, following international alerts and diagnosis of the first cases within the UK, developed molecular sequencing using a conserved region of the genome that provides rapid clade differentiation [12]. To allow for faster diagnosis, an in-house polymerase chain reaction has been developed to rapidly detect *C. auris*. They also demonstrated clade-specific differences in *C. auris* isolate behaviour with two distinct morphological forms—the “clumping” aggregative (S. African) form appearing less pathogenic in an in vivo wax moth model to versus single cell forms [13]. They have also provided antifungal susceptibility testing and expert guidance on all referred isolates, demonstrating that to date no multidrug resistant strains have been found in England. Resistance patterns by clade type can inform choice of drug class administration, and ensure that guidance documents reflect circulating clade epidemiology. To assist diagnostic laboratories in assessing their diagnostic capabilities, a comparison of common routine detection methods has been published [14].

The MRL actively collaborates with novel pharmaceutical development on emerging antifungal agents with activity against *C. auris,* and with universities attached to academic hospitals that have undertaken whole genome sequencing, though its utility is limited given the extremely clonal nature of intra-clade *C. auris* [6]. 

### 3.4. Scientific Output and International Collaboration 

As for any new emerging pathogen, there is a wealth of both rapid information generation and multiple unanswered questions about all aspects of *C. auris* infection. As a training experience opportunity for junior staff, a literature review was performed and a synthesis of all existing data constructed, highlighting the current information on global epidemiology, genomic analysis, identification and typing, cell biology, resistance profiles and treatment, known risks for colonisation and infection, IPC measures, and outbreak costs [3]. This has informed preparation of a wide variety of internal ministerial, parliamentary and government briefings.

The *C. auris* national IMT, affected hospitals and associated academic and scientific institutions continue to improve understanding of the pathogen and add to the international literature through individual and collaborative research projects. For example, staff from PHE’s Biosafety Investigation Unit at Porton Down conducted air sampling in outbreak scenarios, highlighting the potential for *C. auris* to become airborne during aerosol generating high turbulence activities such as bed making [15]. They also investigated the fungicidal activity of a variety of disinfectants and antiseptics commonly used in the UK, with feedback on suggested products received at a national IPC conference [16]. 

Regular discussions with international colleagues including key affected hospital healthcare professionals and sister national public health agencies (particularly calls coordinated by the United States’ Centers for Disease Control and Prevention (CDC)) has been key to learn about other country’s experiences in both successfully dealing with outbreaks, and avoiding potential pitfalls. This has involved regular teleconferencing and engagement at breakout session in international meetings, alongside delivery of multiple international conference presentations to highlight the English landscape. Within Europe, we have provided advice individually to countries with large *C. auris* outbreaks, most notably with Spanish colleagues, participated in pan-European teleconferences coordinated by the he European Centre for Disease Prevention and Control (ECDC), and assisted in regional risk assessments [17].

### 3.5. Professional Engagement 

A key challenge to early recognition of problems with any new pathogen, and preventing onward spread before it takes root in healthcare settings, is awareness among key healthcare professionals that a potential problem exists. Considerable effort was made in England to make relevant parties aware of newly developed guidance detailing local epidemiology, infection prevention and control measures, diagnostic difficulties and clinical management. Alerts were cascaded to all public health and diagnostic laboratories to disseminate onwards to laboratory specialists, microbiologists, infectious diseases physicians and clinical colleagues. Relevant healthcare professionals were informed though national briefing notes and publications in Health Protection Reports [4], a national weekly public health bulletin for England and Wales produced by PHE with widespread distribution across the health sector. Specific *C. auris* updates were included in yearly national “English surveillance programme for antimicrobial utilisation and resistance” (ESPAUR) reports [18], the reference document for national data on antimicrobial prescribing, resistance, and stewardship implementation, education and engagement activities. 

The national team also provided ministerial briefings and responded to media enquiries—these assisted in setting the scene for political and social support for developing a response to novel pathogens, and in providing appropriate reassurance that public health actions were both appropriate and ongoing.

Direct assistance was provided to any newly-affected hospital identified through the triangular surveillance approach. We participated in regional microbiologist network meetings and jointly co-managed events such as organising a half-day training session in the major national IPC conference. Many members of the IMT additionally presented at key national and regional infection conferences. 

## 4. Discussion

At present, England is the only country to implement all recommended surveillance, laboratory, and engagement activities by the European Centre for Disease Control (ECDC) [19]. *C. auris*, as the first nosocomial fungal pathogen to emerge that is easily transmissible from person to person and difficult to eradicate from the environment, has provided unique IPC and outbreak response challenges. The national IMT has developed a series of systems to monitor trends within England, respond to multiple strands of outbreak response and enhance the scientific literature. The national response has also highlighted many challenges, including how to rapidly develop an evidence base to inform both national guidance and individualised and contextualised assistance to specific hospitals or care environments. Key to developing the understanding and limiting transmission of this new pathogen were close communications, networking, and the sharing of information.

There have been considerable challenges with developing a response to *C. auris*: it has uniquely challenging characteristics of persistence, ability to spread, and resistance to general IPC methods; it has unknown population prevalence which renders decision making about screening challenging; there remain large uncertainties about how to best tackle environmental contamination; and there are important future questions about preventing the development of pan-antifungal resistance. 

One important remaining question is how and when to de-escalate a national incident, converting the outbreak response into “business as usual” activities, with normalisation of practice into routine surveillance and reporting while retaining vigilance and the capacity to rapidly respond to new outbreaks. This will be achieved through developing routine automated reports sent to HPTs, highlighting in near real-time any new case detections in a specific geographic region.

The broad aspects of developing a national response to novel pathogens will remain a constant need in the public health sphere, and there has been significant learning from the *C. auris* response. Having detailed “peace-time” plans for advanced preparation for novel nosocomial threats will enhance future outbreak response. For example, this may include producing templates for quick formulation of new guidance documents; developing standardised operating procedures for field epidemiologists and national surveillance centres to capture relevant data to elucidate risk factors; considering how existing surveillance may not address the needs for novel pathogens; updating core contact lists for key professional, governmental, and administrative groups to allow for rapid information distribution; creating the availability of centralised funding for exploratory epidemiological, laboratory and environmental disinfection work; and enabling hospitals or care environments lacking either diagnostic or surveillance capacity to receive dedicated support. 

The arrival of *C. auris*, as well as a serious threat to human health, has provided an opportunity to learn from national public health incident response and inform resilience measures in advance of future outbreaks.

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
