# Peer review of "National Public Health Response to Candida auris in England"

_jof, 2019, doi:10.3390/jof5040093_

Round 1

Reviewer 1 Report

This is a very well written, important paper on the public health response to the emergence of C. auris in England. It provides a nice roadmap for other countries facing similar

One main area that can be added to the paper if room allows is education and outreach. What did PHE do to improve awareness of the pathogen with microbiologists, infection preventionists, etc. That has been a huge part of our response in the U.S., and I imagine much effort was made locally in England to educate relevant stakeholders.

Some minor suggestions:

Line 14: There is now a possible fifth clade of C. auris based on this recent article that came out.  https://wwwnc.cdc.gov/eid/article/25/9/19-0686_article

Line 17: The authors could consider adding a little more information on what types of patients were screened among the 800 screened—all comers? Patients with specific risk factors? This might help other countries considering screening protocols.

Author Response

Reviewer 1

This is a very well written, important paper on the public health response to the emergence of C. auris in England. It provides a nice roadmap for other countries facing similar

One main area that can be added to the paper if room allows is education and outreach. What did PHE do to improve awareness of the pathogen with microbiologists, infection preventionists, etc. That has been a huge part of our response in the U.S., and I imagine much effort was made locally in England to educate relevant stakeholders.

Many thanks for suggesting this – we have added in a new section detailing the steps and channels used for professional engagement, education and outreach (3.v Professional Engagement).

Some minor suggestions:

Line 14: There is now a possible fifth clade of C. auris based on this recent article that came out.  https://wwwnc.cdc.gov/eid/article/25/9/19-0686_article

Thank you for this useful comment - we have added in a line to the effect of a putative fifth clade from Iran, including reference (Line 16).

Line 17: The authors could consider adding a little more information on what types of patients were screened among the 800 screened—all comers? Patients with specific risk factors? This might help other countries considering screening protocols.

Thanks - we have included additional information about the population screened in this section (Lines 96-7).

Reviewer 2 Report

Reviewer comments

The present perspective report from dr. Brown and dr. Guy is extremely well-written and covers the most important parts of the process of handling C. auris in England. The article will definitely be used in understanding the English experience as an inspiration for future work with similar pathogens.

I have only a few questions and suggestions for this meticulously written perspective.

I think the authors need to describe the major challenges in combating C. auris. I believe the information to the public and/or handling the press are important to attract attention and get political and social support for pathogens such as C. auris. I would suggest including a section or a paragraph describing important parts of this process. Looking back, was there anything that could be done in a better way?

And finally, how did the collaboration with other European countries work

Author Response

Reviewer 2

The present perspective report from dr. Brown and dr. Guy is extremely well-written and covers the most important parts of the process of handling C. auris in England. The article will definitely be used in understanding the English experience as an inspiration for future work with similar pathogens.

I have only a few questions and suggestions for this meticulously written perspective.

I think the authors need to describe the major challenges in combating C. auris.

Thank you for this suggestion – we have expanded on this in the discussion section (Lines 186-190).

I believe the information to the public and/or handling the press are important to attract attention and get political and social support for pathogens such as C. auris. I would suggest including a section or a paragraph describing important parts of this process.

Many thanks – we have included this in the new Section 3.v.

Looking back, was there anything that could be done in a better way?

We believe we have included this in the Discussion (Lines 1097-206) but have indicated that these are learning points that have arisen out of the C. auris response.

And finally, how did the collaboration with other European countries work

Thanks, we have added a sentence to the Section 3.iv, Lines 149-152.